# Forest Canopy Height Estimation by Integrating Structural Equation Modeling and Multiple Weighted Regression

Hongbo Zhu [1], Bing Zhang [1,2,*], Weidong Song [1,2], Qinghua Xie [3], Xinyue Chang [1] and Ruishan Zhao [1]

1 School of Geomatics, Liaoning Technical University, Fuxin 123000, China; 47211053@stu.lntu.edu.cn (H.Z.); songweidong@lntu.edu.cn (W.S.); 472120793@stu.lntu.edu.cn (X.C.); zhaoruishan@lntu.edu.cn (R.Z.)
2 Collaborative Innovation Institute of Geospatial Information Service, Liaoning Technical University, Fuxin 123000, China
3 School of Geography and Information Engineering, China University of Geosciences (Wuhan), Wuhan 430074, China; xieqh@cug.edu.cn
* Correspondence: zhangbing@lntu.edu.cn

**Abstract:** As an important component of forest parameters, forest canopy height is of great significance to the study of forest carbon stocks and carbon cycle status. There is an increasing interest in obtaining large-scale forest canopy height quickly and accurately. Therefore, many studies have aimed to address this issue by proposing machine learning models that accurately invert forest canopy height. However, most of the these approaches feature PolSAR observations from a data-driven viewpoint in the feature selection part of the machine learning model, without taking into account the intrinsic mechanisms of PolSAR polarization observation variables. In this work, we evaluated the correlations between eight polarization observation variables, namely, T11, T22, T33, total backscattered power (SPAN), radar vegetation index (RVI), the surface scattering component (Ps), dihedral angle scattering component (Pd), and body scattering component (Pv) of Freeman-Durden three-component decomposition, and the height of the forest canopy. On this basis, a weighted inversion method for determining forest canopy height under the view of structural equation modeling was proposed. In this study, the direct and indirect contributions of the above eight polarization observation variables to the forest canopy height inversion task were estimated based on structural equation modeling. Among them, the indirect contributions were generated by the interactions between the variables and ultimately had an impact on the forest canopy height inversion. In this study, the covariance matrix between polarization variables and forest canopy height was calculated based on structural equation modeling, the weights of the variables were calculated by combining with the Mahalanobis distance, and the weighted inversion of forest canopy height was carried out using PSO-SVR. In this study, some experiments were carried out using three Gaofen-3 satellite (GF-3) images and ICESat-2 forest canopy height data for some forest areas of Gaofeng Ridge, Baisha Lizu Autonomous County, Hainan Province, China. The results showed that T11, T33, and total backscattered power (SPAN) are highly correlated with forest canopy height. In addition, this study showed that determining the weights of different polarization observation variables contributes positively to the accurate estimation of forest canopy height. The forest canopy height-weighted inversion method proposed in this paper was shown to be superior to the multiple regression model, with a 26% improvement in r and a 0.88 m reduction in the root-mean-square error (RMSE).

**Keywords:** forest canopy height; GF-3; ICESat-2; structural equation modeling; PSO-SVR

## 1. Introduction

Forest canopy height is the basic data for carbon stock and carbon cycle analysis of terrestrial ecosystems and one of the important components of global ecological environmental change research. Rapid and accurate acquisition of forest canopy height over a wide area is of great significance when determining the carbon stock and carbon cycle

status of terrestrial forests in a timely and dynamic manner [1–6]. The traditional method of obtaining forest canopy height, as represented by manual forest surveying, is characterized by point-based measurement, which is both time-consuming and laborious. As such, it is difficult to meet the requirement of obtaining forest canopy height data over a wide area with manual forest surveying [7–9].

LiDAR is one of the best data sources for obtaining forest parameter information, as represented by forest canopy height, because of its ability to provide effective observation of the height and spatial distribution of targets on the ground [4,10–12]. In particular, satellite-mounted LiDAR systems are an important data source for the study of global ecological environmental change because of their ability to measure forest structure within the radar footprint at a wide range of scales [13,14]. However, since satellite-based LiDAR systems acquire mostly discontinuous height data, it is still not easy to meet the requirement of acquiring forest canopy height data over a wide range of surface areas [6]. The existing studies have mostly utilized multi-source LiDAR data or methods such as interpolation to meet the needs of acquiring forest canopy height data over a wide range of surface areas [15].

Synthetic aperture radar (SAR) has become an effective remote sensing tool for obtaining the biophysical parameters of forests at regional and global scales due to its all-weather imaging capability and its sensitivity to the physical and geometric properties of the objects of interest. The combination of SAR and LiDAR offers the possibility of accurately detecting the height of forest canopies over a wide area. Forest canopy height estimation methods based on SAR observations can be broadly classified into backscatter model–based methods, interferometric synthetic aperture radar (InSAR)–based methods, polarimetric InSAR (PolInSAR)–based methods, and data-driven empirical modeling–based methods [16–20].

SAR systems mainly estimate forest biomass based on the backward scattering coefficients, but the backward scattering coefficients of different wavelengths and polarization modes are obtained from different parts of the tree, which makes the estimation of forest canopy height difficult [21,22]. A number of backscatter modeling methods have been developed [23,24], most of which are based on first-order or second-order solutions for the radiative transfer equation [25]. These models perform better in the estimation of forest canopy height in the co-polarization mode, but the accuracy of cross-polarized forest canopy height estimation tends to be reduced because these models ignore the higher-order solution for the radiative transfer equation [26,27].

The data-driven empirical modeling approach is a method that can be used for large-scale vegetation height estimation based on the use of machine learning regression to train an estimation model between the PolSAR observations and canopy height. The trained regression model is then used to estimate the forest height in the study area with the corresponding PolSAR observations [28,29]. Since this method uses polarization observation information, its accuracy is usually superior to forest height estimation based on backscatter coefficients. Currently, there are more studies using PolSAR observations to invert forest canopy height based on data-driven ideas. However, most of these studies are based on the linear correlation between PolSAR observations and forest canopy height, a priori knowledge, or multiple adjustments of parameters to select the optimal combination of features [30–34]. In addition, empirical relational models or linear/polynomial regression models are typically chosen because of the limited number and type of PolSAR observations [16,35]. However, the above methods only perform feature selection of PolSAR observations from a data-driven perspective and do not take into account the intrinsic mechanisms between the PolSAR polarimetric decomposition features and the backscattering coefficients.

Structural equation modeling (SEM) is an analytical method that empirically determines causality based on correlation [36,37]. Using SEM, the composite causal relationship between PolSAR decomposition features, backscattering coefficients, and forest canopy height can be analyzed. Therefore, in this paper, we propose a method for the weighted estimation of forest canopy height based on the above composite causality, combined with

a machine learning regression model. Thus, to analyze the contribution of the different variables to the estimation of forest canopy height, a machine learning model was constructed to accurately estimate the forest canopy height. Then, based on the multiple weighted regression method, the forest canopy height estimation was conducted.

To solve the problem of the feature selection in the current PolSAR forest canopy height estimation task combined with machine learning being data-driven and the fact that it does not take into account the intrinsic mechanism between polarimetric observation variables, we propose a forest canopy height estimation method based on SEM that takes into account the relationship between polarimetric observation variables. The objectives of this study were: (1) to analyze the contribution of the different polarimetric observation variables to forest canopy height estimation; (2) to evaluate the effectiveness of the contribution of the different variables to forest canopy height estimation from various perspectives, including individual effectiveness and interactions; and (3) to construct a machine learning model for accurately estimating the height of forest canopies over a wide range of scales.

The remainder of this paper is organized as follows. Section 2 describes the study area and data preparation. Section 3 explains the research methodology, including the forest canopy height estimation weight calculation method combining SEM with the Mahalanobis distance, and the support vector regression model based on particle swarm optimization (PSO-SVR). Section 4 presents the experimental results. Section 5 discusses the validity of single polarimetric observation variables in the estimation of forest canopy height, provides an analysis of the coupled contribution of multiple polarimetric observation variables, and discusses the limitations of the proposed approach and potential future research. Section 6 concludes and summarizes the current work.

## 2. Materials and Methods

### 2.1. Study Area and Data

As shown in Figure 1, we selected a part of the forested area of Gaofeng Ridge in Baisha Lizu Autonomous County, Hainan Province, China, which is suitable for forest growth due to the abundant precipitation and high temperature occurring in the same season [38]. The topography of the study area is predominantly mountainous and hilly, with a variety of land-cover types, including forest, water bodies, and buildings. The vegetation in the study area is mostly tropical or subtropical trees, and the height of the vegetation is about 0–40 m [39]. It is noteworthy that the typical range of forest canopy height in the study area is 10–25 m [39]. The area delineated by the red rectangle in Figure 1a is the extent of the three Gaofen-3 (GF-3) fully polarized SAR images covering the study area acquired on 5 January 2021, and the blue dots mark all the experimental data used for the estimation of forest canopy height. We performed the validation of the inversion method proposed in this paper in the region marked by the blue box. Figure 1b shows the Sentinel-2 image obtained in this region on 2 January 2021. Sentinel-2 is a multispectral imaging sensor operated by the European Space Agency and provides open, freely accessible data (https://dataspace.copernicus.eu/, accessed on 28 June 2023). We divided the training and test sets randomly in the ratio of 0.9:0.1 for the training and accuracy evaluation of the forest canopy height estimation [40]. The land and vegetation height (ATL08) product is derived from publicly available data from the National Snow and Ice Data Center (https://nsidc.org/data, accessed on 17 April 2023) [41–43]. The data were obtained from a 17-m diameter footprint acquired by a laser pulse. The sampling density of neighboring spots was 0.7 m [44,45]. In this study, the relative canopy height was chosen as the reference value of forest canopy height [46–48].

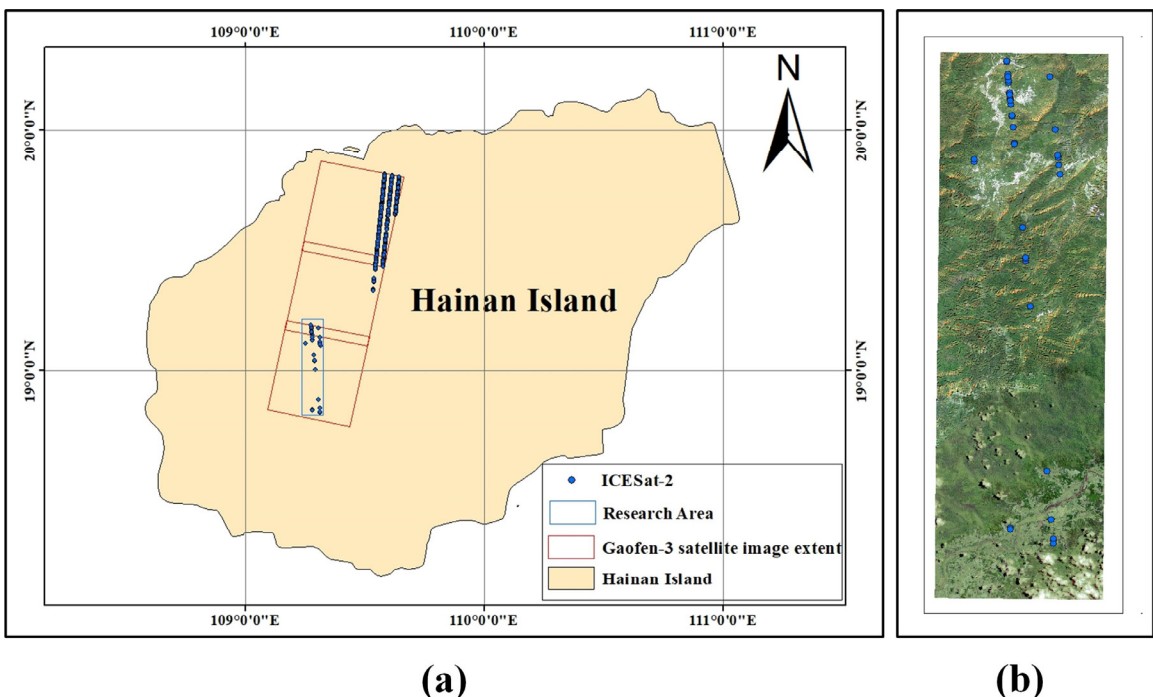

**Figure 1.** Map of the study area. (**a**) GF-3 satellite image extent; (**b**) Sentinel-2 image.

The ICESat-2 LiDAR satellite measurements used for the ALT08 canopy height information and the GF-3 data were not acquired at the same time due to the different satellite sampling times. However, since the maximum time offset was 263 days (occurring on 25 September 2021), it is reasonable to assume that forest heights in the study area have not changed significantly over this time period.

At the same time, a total of three consecutive Gaofen-3 satellite (GF-3) QPS1 fully polarized data scenes acquired in 2021 were used in this study. Table 1 lists the acquisition date, center latitude and longitude, resolution, and imaging mode information for the three GF-3 images.

**Table 1.** Details of the GF-3 images acquired in 2021.

| Date | Mode | Center Longitude | Center Latitude | Resolution |
|---|---|---|---|---|
| 05 January 2021 | QPS1 | 109.3° E | 19.0° N | 8.0 m × 8.0 m |
| 05 January 2021 | QPS1 | 109.4° E | 19.3° N | 8.0 m × 8.0 m |
| 05 January 2021 | QPS1 | 109.5° E | 19.6° N | 8.0 m × 8.0 m |

The ICESat-2 LiDAR satellite measurements used for the ALT08 canopy height information and the GF-3 data were not acquired at the same time, due to the different satellite overpass times. However, since the maximum time offset was 263 days (occurring on 25 September 2021), it is reasonable to assume that there was no significant change in forest canopy height and its trend in the study area between the date of acquisition of the GF-3 microwave radar imagery and the date of the ICESat-2 acquisition of the ground canopy height.

We performed a series of pre-processing operations on each GF-3 image, including calibration, filtering, and geocoding. The 30-m spatial resolution provincial digital elevation model (PDEM) of Hainan Province, China, was used to geocode each scene image into the Universal Transverse Mercator (UTM) coordinate system. The final pixel width of the geocoded product was 10 m. Due to the complex and varied topography of the study area, the relief of the terrain can have an impact on the polarization observations. In order to

eliminate the polarization observation errors caused by the terrain, we performed terrain radiometric correction on the three GF-3 images.

### 2.2. Polarimetric Synthetic Aperture Radar Observations

The second-order scattering matrix (S-matrix), which is the most commonly used data representation in PolSAR systems, contains the energy features, phase features, and polarimetric features of the feature target, as shown in Equation (1) [49,50]:

$$S = \begin{bmatrix} S_{HH} & S_{HV} \\ S_{VH} & S_{VV} \end{bmatrix} \tag{1}$$

where the $S_{HV}$ element represents the information of the scattered echo in the vertical direction acquired by the satellite antenna after the fully polarized SAR radar wave is emitted horizontally and polarized with the ground object. $H$ and $V$ represent the horizontal and vertical directions, respectively. This can be understood in the same way as the physical significance of the other parameters in Equation (1). The second-order scattering matrix $S_{HH}$, $S_{HV}$, $S_{VH}$, $S_{VV}$ elements are complex scattering parameters, where each element is in the form of a complex number and contains the amplitude and phase information of the scattering information. In the process of PolSAR data processing, Sinclair scattering matrix vectorization is typically used to obtain the scattering vector $k$ of the target, as shown in Equation (2), where the superscript $T$ is the matrix transpose symbol. In addition, according to the reciprocity theorem [45,46], for the backward scattering of PolSAR data, where the scattering matrix is a symmetric matrix, the three-dimensional Pauli basis vector $k_p$ and lexicographic basis vector $k_l$ are as shown in Equations (3) and (4).

$$k = [S_{HH}, S_{HV}, S_{VH}, S_{VV}]^T \tag{2}$$

$$k_p = \frac{1}{\sqrt{2}}[S_{HH} + S_{VV}, S_{HH} - S_{VV}, 2S_{HV}]^T \tag{3}$$

$$k_l = \left[S_{HH}, \sqrt{2}S_{HV}, S_{VV}\right]^T \tag{4}$$

The third-order Pauli basis coherence matrix $T$ can be obtained by multiplying the three-dimensional Pauli basis vector $k_p$ and the target scattering vector of the lexicographic target basis vector with their own conjugate transpose in an outer product, as shown in Equation (5) [51,52], where $(.)^*$ is the conjugate complex number.

$$T = \left\langle k_p \cdot k_p^T \right\rangle = \begin{bmatrix} T11 & T12 & T13 \\ T21 & T22 & T23 \\ T31 & T32 & T33 \end{bmatrix} = \frac{1}{2} \begin{bmatrix} \left\langle |S_{HH} + S_{VV}|^2 \right\rangle & \left\langle (S_{HH} + S_{VV})(S_{HH} - S_{VV})^* \right\rangle & 2\left\langle (S_{HH} + S_{VV})S_{HV}^* \right\rangle \\ \left\langle (S_{HH} - S_{VV})(S_{HH} + S_{VV})^* \right\rangle & \left\langle |S_{HH} - S_{VV}|^2 \right\rangle & 2\left\langle (S_{HH} - S_{VV})S_{HV}^* \right\rangle \\ 2\left\langle S_{HV}(S_{HH} + S_{VV})^* \right\rangle & 2\left\langle S_{HV}(S_{HH} - S_{VV})^* \right\rangle & 4\left\langle |S_{HV}|^2 \right\rangle \end{bmatrix} \tag{5}$$

Based on the above polarimetric observation matrix, it is possible to obtain multiple types of polarimetric parameters for forest canopy height estimation [51–54]. Eight polarimetric observation variables were selected in this study (see Table 2).

**Table 2.** List of the eight polarimetric observables selected in this study.

| Polarimetric Observation Variable | Description |
|:---:|:---:|
| T11, T22, T33 | Backscattering coefficients in the Pauli polarization channels |
| SPAN | Total backscattered power |
| Ps, Pd, Pv | Scattering power from the different scattering mechanisms derived from Freeman-Durden decomposition |
| RVI | Radar vegetation index |

Firstly, we considered the diagonal elements T11, T22, and T33 in the matrix of polarimetric observation variables. Secondly, we selected the total backscatter power (i.e., the SPAN), which can be extracted based on the diagonal elements in the sum of the coherence matrix. The common method of obtaining polarimetric features is the use of polarimetric target decomposition. This, in turn, can be classified into coherent polarimetric decomposition and non-coherent decomposition [49,50]. Non-coherent polarimetric decomposition is more suitable for use with natural targets, due to its ability to provide a distributed description of the target features [49,50]. Non-coherent polarimetric decomposition can be further classified into model-based decomposition and decomposition based on eigenvector/eigenvalue analysis. Freeman-Durden three-component decomposition is one of the earliest and most popular of the model-based decomposition methods and produces three scattering power parameters representing the mechanisms of surface scattering, dihedral angle scattering, and body scattering (Freeman and Durden, 1998) [55]. These parameters are the three scattering power parameters selected in this study.

The radar vegetation index (RVI) [56] is sensitive to forest morphological characteristics, so it was also considered in this study, as shown in Equation (6), where $\sigma HV$ is the cross-polarized backward scattering coefficient, and $\sigma HH$ and $\sigma VV$ are the co-polarized backward scattering coefficients expressed in power units.

$$RVI = \frac{8\sigma HV}{\sigma HH + \sigma VV + 2\sigma HV} \tag{6}$$

Moreover, we obtained a total of 553 LiDAR altimetry sampling points within the coverage of the GF-3 imagery in the study area, based on the geographic location records of the sampling points of the ALT08 land and vegetation height product of ICESat-2. We then combined the eight polarimetric observation variables to construct a forest canopy height estimation dataset for the training of the forest canopy height estimation models and the accuracy validation.

## 3. Research Methodology

The PolSAR forest canopy height estimation method based on the structural equation model (SEM) proposed in this paper consists of a weight calculation module that combines the structural equation model (SEM) with the Mahalanobis distance and a forest height estimation module, as shown in Figure 2. The weight calculation module based on SEM calculates the contribution of each polarimetric observation variable to the forest canopy height estimation from the input PolSAR data and calculates the weight of each polarimetric observation variable using the Mahalanobis distance. The forest height estimation module is an SVR model based on PSO (PSO-SVR), which is used to perform multivariate weighted estimation of the forest canopy height and derive the optimal parameters.

### 3.1. Weight Calculation Based on Structural Equation Modeling (SEM)

SEM is a multivariate statistical technique method used to explain the causal and correlational relationships between variables [57–59]. Compared with traditional multivariate statistical methods, SEM allows the measurement of latent variables and the construction of path diagram models, which provides the possibility of analyzing the complex coupled relationships between multiple independent variables and the target dependent variable. As a result, SEM has been widely used in the fields of ecology [60–62] and forestry [63].

The PolSAR-based forest canopy height estimation mechanism we developed is complex and is in the presence of multiple causal and correlational relationships, such as different polarization modes and polarimetric decompositions. Therefore, in addition to considering the direct contribution of each of the multiple polarimetric parameters to the forest canopy height estimation, we also needed to consider the indirect contribution of the interactions among the variables to forest canopy height estimation. The core of SEM analysis is that it allows all the parameters in the model to be estimated simultaneously by the maximum likelihood estimation method. It also allows the degree of model fit as

a whole to be determined through the difference between the theoretical and the actual measured covariance. The structural equation model obtains the path coefficients of the model by calculating the covariance matrices between the polarization mode, polarimetric decomposition, and forest canopy height. These can then be used to quantitatively characterize the mechanism of the interaction between the multiple parameters and forest canopy height. The path coefficients derived from the model results are used to characterize the contribution of the different polarimetric observation variables to the estimation of the forest canopy height and are used for determining the weights of the forest canopy height estimation model.

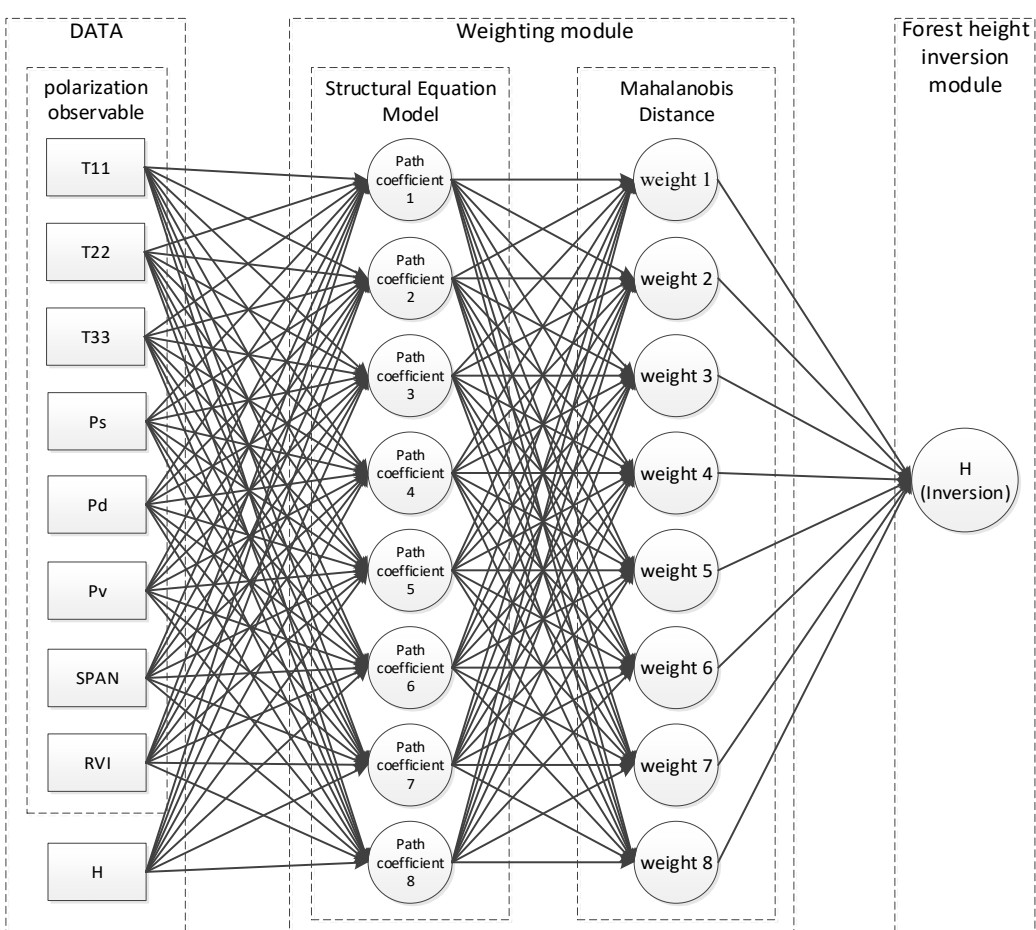

**Figure 2.** Structure of the forest canopy height estimation model.

Variables in SEM can be divided into observed variants and latent variables, where the observed variants are variables that can be directly obtained and measured, and the latent variables are variables that cannot be directly obtained or characterized. The polarization mode and polarimetric decomposition parameters described in Sections 2.1 and 2.2 of this paper can be directly obtained and characterized, so the structural equation model used in this study uses only the observed variables as model inputs to explore the relationships between the eight parameters and forest canopy height. In forest canopy height estimation, since the different variables contribute to the estimation results to different degrees, applying different weights to them can highlight the data characteristics and improve the estimation accuracy, to some extent [64]. Based on this, we propose to determine the weights between the different polarimetric observation variables and forest canopy height based on SEM, while taking into account both the direct and indirect contributions of the different variables to the estimation of forest canopy height, and combining the path coefficients of SEM. Aiming at the characteristics of the polarimetric observation variables, such as the large-scale difference and complex correlation, the Mahalanobis distance is used to calculate the

above weights. The Mahalanobis distance is not affected by the spatial dimension of the data features and can effectively eliminate the interference of correlation between variables. Therefore, in this study, we used the Mahalanobis distance between each polarimetric observation variable and the ICESat-2 forest canopy height [65] as the basis for the weight calculation.

$$D_M(x,y) = \sqrt{(x-y)^T \sum^{-1} (x-y)} \tag{7}$$

The Mahalanobis distance is calculated as shown in Equation (7), where $\sum^{-1}$ is the inverse matrix of the covariance matrix between the variables, which is calculated by SEM, as shown in Equation (8), where y is the forest canopy height, $x_n$ is the polarimetric observation variables, var is the variance of the variables, and cov is the covariance between the variables. The weight calculation module first calculates the Mahalanobis distance of each variable from the ICESat-2 forest canopy height based on the covariance matrix of SEM and then calculates the weights.

$$\sum = \begin{bmatrix} var(y) \\ cov(x_1,y) & var(x_1) \\ cov(x_2,y) & cov(x_2,x_1) & var(x_2) \\ cov(x_3,y) & cov(x_3,x_1) & cov(x_3,x_2) & \ddots \\ \vdots & \vdots & \vdots & & var(x_{n-1}) \\ cov(x_n,y) & cov(x_n,x_1) & cov(x_n,x_2) & & cov(x_n,x_{n-1}) & var(x_n) \end{bmatrix} \tag{8}$$

### 3.2. Support Vector Regression Model Based on Particle Swarm Optimization (PSO-SVR)

Machine learning–based methods are commonly used for classification and regression problems in remote sensing due to their strong predictive ability [66,67]. Among the different methods, SVR is a supervised machine learning algorithm based on the kernel function, which takes into account the global nature of parameters. SVR and its improved versions are commonly used in parameter estimation based on remote sensing imagery because of the fast convergence, few parameters, and high reliability [68–70]. Due to the complex and nonlinear relationship between the polarimetric observation variables and the forest canopy height, we adopted PSO to automatically search for the key hyperparameters in the SVR model, which can effectively avoid the occurrence of overfitting and underfitting. This method is similar to the genetic algorithms, which use the population fitness to determine the optimal solution for a problem, and it can be used for the optimization of nonlinear problems [71,72]. The PSO algorithm was a new method for providing optimal solutions [73]. This approach avoids repetitive parameter adjustments and enables fast access to combinations of hyperparameters with global optimization as well as estimation results. Notably, it also prevents the results from falling into local optimality. In PSO, each particle records and updates its velocity, position, and the optimal distance between itself and the swarm. For an optimization problem in an *n*-dimensional space with *m* particles forming the population, the position of the *i*th particle is denoted as *xi* = (*x*1, *x*2, . . ., *xn*) and the velocity is denoted as *vi* = (*v*1, *v*2, . . ., *vn*). The formula for updating the velocity and position of each particle is as follows:

$$v_i(t+1) = \omega v_i(t) + c_1 r_1(p_i(t) - x_i(t)) + c_2 r_2(p_g(t) - x_i(t)) \tag{9}$$

$$x_i(t+1) = x_i(t) + v_i(t+1) \tag{10}$$

where $i = 1, 2, 3, . . ., m$; $p_i$ is the location of the individual extreme point; $p_g$ is the location of the global extreme point; $\omega$ is the initial value of the inertia weight; $c_1$ and $c_2$ are the acceleration coefficients; and $r_1$ and $r_2$ are random numbers between 0 and 1. For the random forest regression model, the inclusion of a PSO algorithm can help to ensure the rationality of the parameter optimization [74]. The PSO algorithm is used to find the

optimal parameters that allow the SVR model to converge and avoid model overfitting and underfitting.

## 4. Experiments and Results

### 4.1. Results of the Weighting Calculations

We combined SEM to construct the Mahalanobis distance between the forest canopy height and polarimetric observation variables, so as to calculate the weights of the observations for the forest canopy height estimation. The input variables to the model are the eight polarized observations described previously. We constructed a path diagram model between the forest canopy height (H) and the eight polarimetric observation variables, based on SEM, as shown in Figure 3. In the path diagram, the polarimetric observation variables are represented by the rectangular boxes, and two variables are connected with a solid line with a single arrow to indicate a causal relationship. Two variables are connected with a dashed line with an arrow to indicate a small causal relationship. In addition, a red line is used to connect the forest canopy height (H) with the polarimetric observation variable, so as to indicate that this variable makes a direct contribution to the forest canopy height estimation. A blue line is used to connect two variables to indicate that they do not make a direct contribution to the forest canopy height estimation but can make an indirect contribution in the forest canopy height estimation through the other polarimetric observation variables. The structural equation model was fitted with a total of 553 sampling points, and the fitted model had a chi-squared value of 0.347 and degrees of freedom (df) of 7. The model fit was evaluated, and the identified model structure was used to reflect the complex relationships between the eight polarimetric observation variables and forest canopy height. The structural equation model is based on the partial least squares (PLS) method for calculating path coefficients, as shown in Figure 3. The high path coefficients between the SPAN and forest canopy height imply that this contributes the most in the forest canopy height estimation. The integrated path coefficients from each polarimetric observation variable to the forest canopy height were calculated from their direct and indirect path coefficients. The results are listed in Table 3. In terms of the integrated path coefficients, T11, T33, and the SPAN are the factors that make a major contribution to the forest canopy height estimation. The weight of each variable was calculated based on the contribution of the polarimetric observation variable to the forest canopy height estimation based on the Mahalanobis distance as depicted in Table 3.

### 4.2. Forest Canopy Height-Weighted Estimation Results

In this study, we input the weights of eight polarization observation variables into the PSO-SVR model based on the weighting method of SEM and used the PSO algorithm to adaptively search for the optimal hyperparameter combinations of the SVR model for the estimation of forest canopy height. For the model parameters, the penalty factor (C) was 2.81, the fault tolerance factor (epsilon) was 0.01, and the coefficient of the kernel function (gamma) was 0.9486. The PSO-SVR model contributed to the forest canopy height estimation by considering the low-correlation variables in addition to the polarimetric observation variables with a high correlation. Compared with the data-driven estimation method that selects polarimetric observation variables with a high correlation, the method proposed in this paper is based on the relevant physical a priori information. This approach is closer to the perception of forest canopy height by PolSAR in the natural environment and integrates a variety of polarimetric observation variables to accurately invert the forest canopy height. From the results, the forest height-weighted estimation results show a lower root-mean-square error (RMSE) and a higher Pearson correlation coefficient (*r*). Compared with the PSO-SVR model, the *r*-value of the forest height-weighted estimation model based on the proposed structural equation model shows an improvement of 26%, with the RMSE reduced by 0.88 m. Figures 4 and 5 show the scatter plots of the forest canopy height estimation results obtained with and without the weighting, where the weighted estimation has the best overall accuracy and a higher correlation compared to the unweighted model.

As expected, the higher accuracy of the weighted forest canopy height estimation results can be attributed to the more reasonable input variable weights.

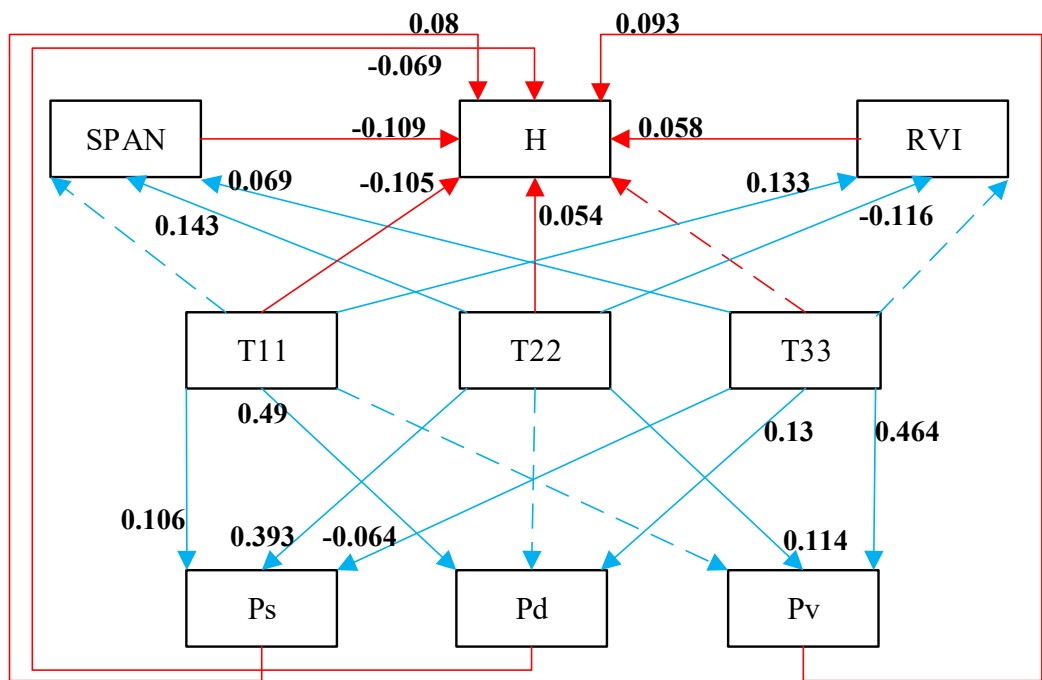

**Figure 3.** Path diagram of the polarimetric observation variables and forest canopy height structural equation modeling.

**Table 3.** Weights of the eight polarimetric observation variables related to forest canopy height.

| Polarimetric Observation Variable | Path Coefficient | Weight |
|---|---|---|
| T11 | 0.163 | 1.144 |
| T22 | 0.101 | 0.581 |
| T33 | 0.11 | 3.766 |
| SPAN | 0.109 | 2.067 |
| RVI | 0.058 | 2.774 |
| Ps | 0.08 | 1.306 |
| Pd | $-0.069$ | 2.111 |
| Pv | 0.093 | 1 |

We used the three GF-3 images and the 553 forest canopy sample points to weight the inversion of forest canopy height in the study area within the scope of Figure 1b, as shown in Figure 6. The black part of the figure shows the non-forested areas consisting of towns, water bodies, and farmland, which we classified as non-forested areas not participating in the forest canopy height inversion, based on expert visual interpretation. From Figure 6, it is apparent that the areas with higher forest canopy heights are mostly concentrated in the densely forested mountainous areas. This is due to the fact that mountainous areas are basically unaffected by anthropogenic activities, and the vegetation is better protected. The height of the forest in the figure is mostly in the range of 16–30 m, and the height of the forest is staggered, which is more in line with the actual structure of forested areas. Low-vegetation areas are mostly located near towns and cities, where human activities are more frequent and vegetation heights are low.

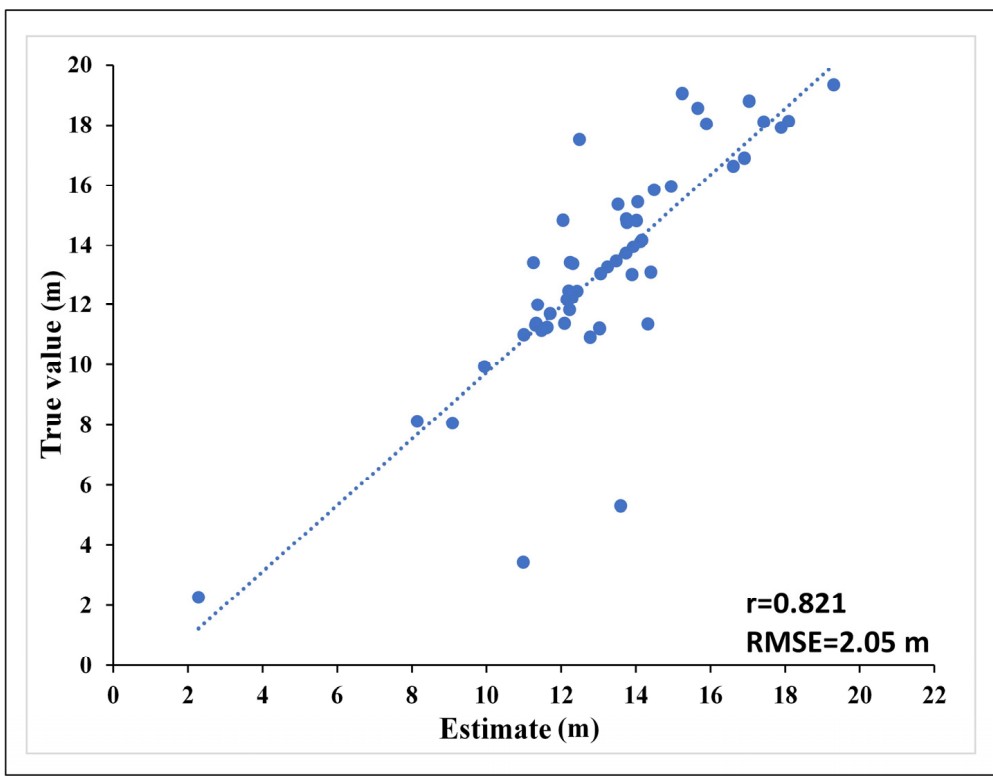

**Figure 4.** The estimation accuracy obtained from the forest height-weighted estimation model in SEM view proposed in this paper.

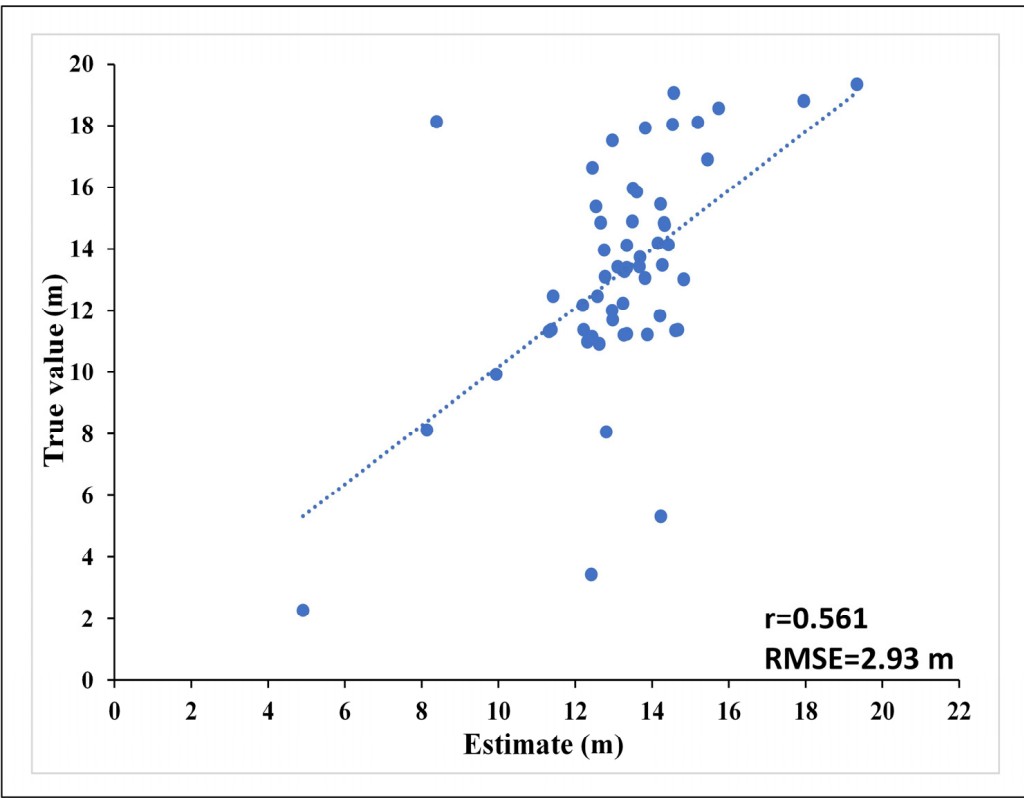

**Figure 5.** The forest height estimation accuracy based on the PSO-SVR model under the same parameter settings.

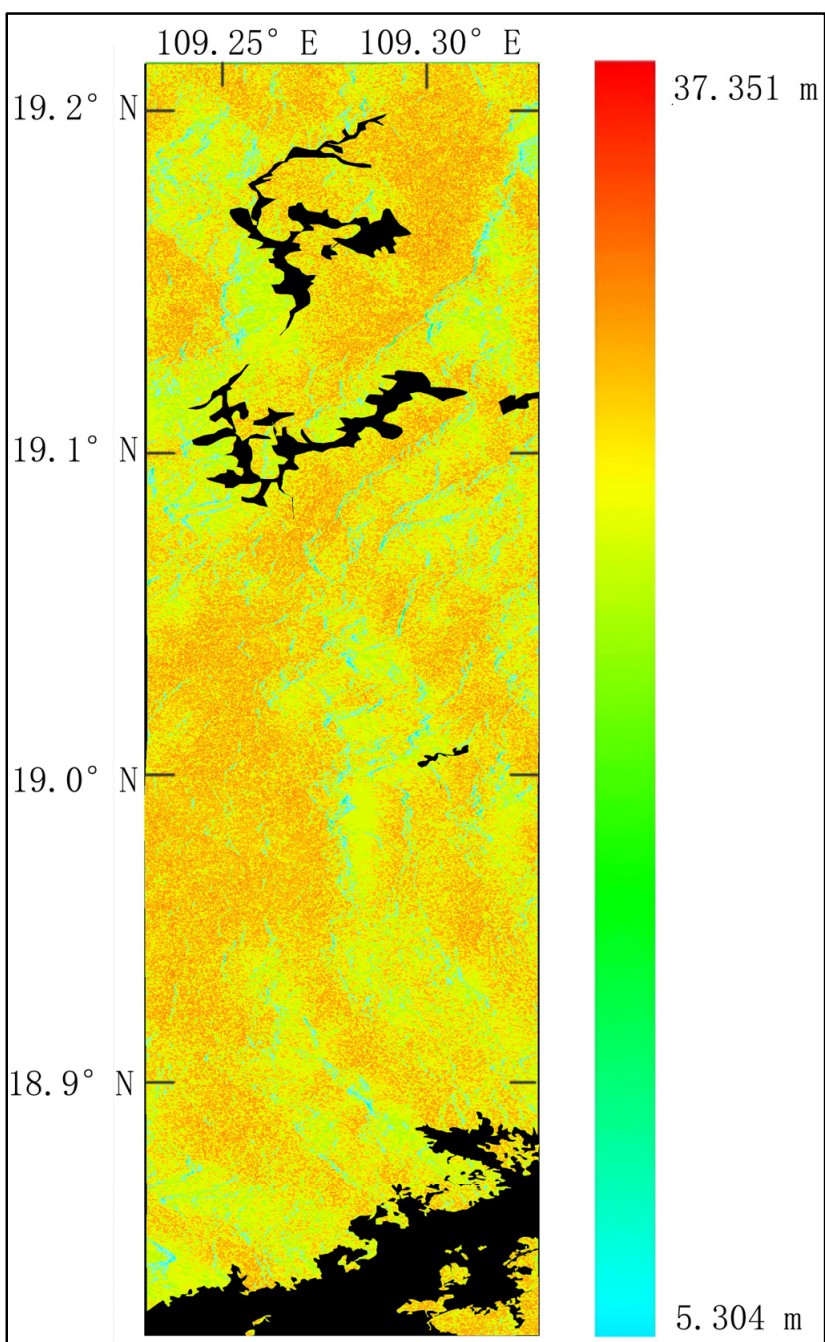

**Figure 6.** Forest canopy height distribution on 5 January 2021, based on the inversion of the forest canopy height-weighted estimation method in the SEM field of view proposed in this paper.

## 5. Discussion

### 5.1. Validity of Single Polarimetric Observation Variables

The Pearson correlation coefficient ($r$) is commonly used to measure the correlation between two variables and has been widely used in various research fields [75]. Therefore, the $r$-value was used as an evaluation index to analyze the correlation between each polarimetric observation variable and the forest canopy height obtained from ICESat-2. In this study, eight input polarimetric observation variables were selected, and their correlations with forest canopy height are shown in Figure 7.

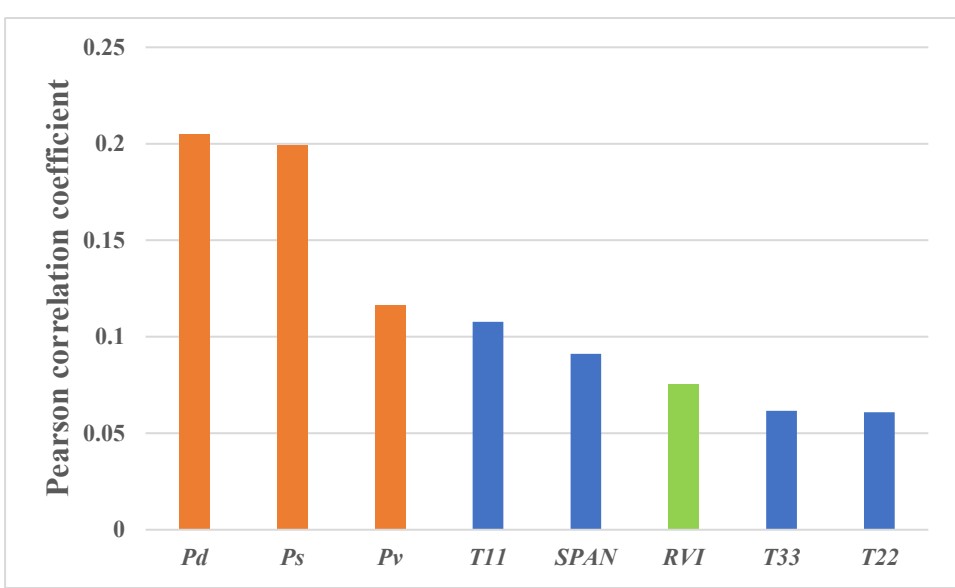

**Figure 7.** Correlations between the polarimetric observation variables and forest canopy height. The different types of polarimetric observation variables are depicted as follows: backscattering coefficients and SPAN (blue), Freeman-Durden decomposition (orange), and RVI (green).

In Figure 7, for a better visual effect, the variables belonging to the same or similar groups are marked with the same color. The correlation between the three scattering components of Freeman-Durden decomposition and forest canopy height is relatively strong compared to the other polarimetric observation variables. Among the different variables, the contribution of the dihedral angle scattering component (Pd) of Freeman-Durden decomposition is the greatest, and the results show a strong correlation between this variable and forest canopy height. The surface scattering component (Ps) of Freeman-Durden decomposition takes second place in the variable importance ranking after Pd. T11 is the fourth most important variable and is highly correlated with the body scattering component (Pv), which ranks third. The SPAN ranks below T11, but is still important, and the radar vegetation index (RVI), T33, and T22 are located at the bottom of the importance list. It is clear from the above analysis that the SAR parameters associated with Freeman-Durden decomposition are highly sensitive to forest canopy height. In addition, in the case of some of these factors not being available, these results provide guidance for the selection and ranking of variables for forest canopy height estimation models. The Pearson correlation coefficient ($r$) only illustrates the correlation between each polarimetric observation variable and the forest canopy height, and it is difficult to differentiate whether this correlation is due to the direct contribution of the observed variable to the forest canopy height or the interaction among variables.

### 5.2. Analysis of the Coupled Contribution of Multiple Polarimetric Observation Variables

The Pearson correlation coefficient ($r$) only describes the linear correlation between the forest canopy height obtained by ICESat-2 and any of the eight polarimetric observation variables, while ignoring the coupled contribution of two or more polarimetric observation variables to the forest canopy height estimation. To quantitatively evaluate the coupling between the multivariate variables and forest canopy height, we constructed a path diagram model between the forest canopy height and the eight different polarimetric observation variables, as shown in Figure 3. Each polarimetric observation variable has a direct effect on the forest canopy height. Since the SPAN is obtained by summing the three diagonal elements (T11, T22, and T33) in the polarimetric observation matrix, T11, T22, and T33 have a direct effect on the SPAN. Moreover, Freeman-Durden three-component decomposition is conducted based on the polarimetric observation matrix (T), so that T11, T22, and T33 have a direct effect on the surface scattering component (Ps), dihedral angle scattering

component (Pd), and body scattering component (Pv). Of particular note, the RVI is shown in Equation (6). The radar vegetation index (RVI) ranges from 0 to 1 and is a measure of the randomness of the scattering. For smooth and light surfaces, the RVI will be close to zero and will increase with increased vegetation growth. Therefore, we consider that T11, T22, and T33 all have an effect on the RVI.

Based on SEM, we quantitatively estimated the extent of the contribution of the above eight polarimetric observation variables to the forest canopy height, which is composed of two parts: direct contribution and indirect contribution. The direct contribution is the value of a single polarimetric observation variable pointing directly to the forest canopy height through a single arrow, as indicated by the red arrow in Figure 3, where all of the eight polarimetric observation variables contribute directly to the forest canopy height estimation task. Indirect contributions are path coefficients that indirectly contribute to the forest canopy height estimation because one polarimetric observation variable has a direct effect on one or more of the other polarimetric observation variables and ultimately acts on the forest canopy height estimation. For example, the combination of blue arrows and red arrows in Figure 3 is the indirect path of the polarimetric observation variables. For instance, the indirect path coefficient of T22 indirectly contributing to the forest canopy height (H) via the SPAN is $0.143 \times 0.109 = 0.0156$. The composite path coefficient is the sum of all the direct and indirect path coefficients for a single polarimetric observation variable. In terms of the combined path coefficients, T11, T33, and the SPAN are the variables that make a major contribution to the forest canopy height estimation, as shown in Table 1. The path coefficients listed in Table 1 differ from the results of the Pearson correlation coefficients (*r*) in Figure 5, which is the result of the interactions between these polarimetric observation variables.

*5.3. Limitations and Future Research*

In this study, we selected the PSO-SVR model to weight the estimation of forest canopy height in the study area using eight polarimetric observation variables: the three diagonal elements (T11, T22, and T33) in the PolSAR polarimetric observation matrix, the three components of Freeman-Durden three-component decomposition (Ps, Pd, and Pv), the SPAN, and the radar vegetation index (RVI). Since the different polarimetric observation variables contribute differently to the forest canopy height estimation, we determined the weights of the polarimetric observation variables for the forest canopy height by combining SEM and the Mahalanobis distance. The results are listed in Table 1. The purpose of the PSO-SVR model is to determine the optimal hyperplane, so that the point furthest away from the hyperplane will have the shortest distance to the plane. In a previously reported study, the effectiveness of the Mahalanobis distance in the PSO-SVR-weighted regression task was demonstrated [56]. Moreover, in this paper, we have proposed a machine learning-based weighted estimation method for the estimation of forest canopy height using GF-3 polarimetric observations, which still has some limitations. Firstly, the method relies on a large volume of available LiDAR altimetry data and remote sensing data. However, although the number of samples was large, there were few samples with a lower forest canopy height. This may have affected the accuracy of both the model calibration and estimation. In addition, the natural conditions, such as soil moisture and vegetation foliar water content, may have affected the estimates. It is worth noting that the density of the forest and the type of trees also have some influence on the results of the forest canopy height estimation. In our future work, we will focus on testing the proposed approach for the height estimation of other vegetation types, such as planted forests and crops. We propose to use principal component analysis for height estimation with different vegetation types and densities. The use of different radar frequencies (e.g., ALOS and LT-1 in the L-band) as well as PolSAR data from P-band airborne radar at different locations will also be investigated for testing and analysis in future studies.

## 6. Conclusions

In this paper, we have described how eight polarimetric observation variables sensitive to forest canopy height in PolSAR data can be selected to analyze their respective and coupled contributions to forest canopy height estimation. Based on this, the PSO-SVR model was used to realize forest canopy height estimation in the study area. The correlation between individual polarimetric observation variables and forest canopy height was first estimated by Pearson correlation analysis, and the coupling correlation between the polarimetric observation variables and forest canopy height was quantitatively estimated based on SEM. The results showed that the correlations between T11, T33, and the SPAN with forest canopy height were larger than those for the other polarimetric observation variables. Secondly, based on the degree of correlation between the eight polarimetric observation variables and forest canopy height, the weight of each polarimetric observation variable in the forest canopy height estimation task was calculated by introducing the Mahalanobis distance. It was found that the weakly correlated variables are also important for estimating forest canopy height as well as improving the estimation accuracy, and the coupling effect of the different polarimetric observation variables can further improve the estimation accuracy. The results showed that the proposed forest canopy height-weighted estimation method combining SEM and the Mahalanobis distance can achieve a higher estimation accuracy than the multiple weighted regression model. In terms of $r$, there was a 26% improvement when compared with multiple weighted regression, and the RMSE showed a 0.88 m improvement. Moreover, the forest canopy height results obtained in the study area are similar to the results of Liu et al. [39]. Compared with previous studies using SAR data for the estimation of forest canopy height, the proposed method represents a promising approach for the utilization of PolSAR data.

**Author Contributions:** Conceptualization, H.Z.; methodology, H.Z., B.Z. and W.S.; software, H.Z. and X.C.; validation, H.Z.; writing—original draft preparation, B.Z.; writing—review and editing, H.Z., W.S. and Q.X.; visualization, H.Z. and R.Z.; project administration, H.Z. and B.Z.; and funding acquisition, W.S. and B.Z. All authors have read and agreed to the published version of the manuscript.

**Funding:** This work was supported in part by the National Natural Science Foundation of China under Grants 42204031 and 42071343 and a project funded by the China Postdoctoral Science Foundation: 2022MD723791.

**Data Availability Statement:** The dataset can be obtained on request by e-mail from https://ciigis.lntu.edu.cn/ (accessed on 3 January 2024).

**Conflicts of Interest:** All authors declare that they have no conflicts of interest.

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
