# Peer review of "Forest Canopy Height Estimation by Integrating Structural Equation Modeling and Multiple Weighted Regression"

_forests, doi:10.3390/f15020369_

Round 1
Reviewer 1 Report
Comments and Suggestions for Authors
The paper is generally well-written and explained - more so than some papers I have read where a complex approach needs to be made accessible to the readership. However, it does need some final editing to remove wordy sentences and sentences that could be better expressed. For example, in the abstract - the first sentence is not the best start! Oriented to the forest carbon stock and carbon cycle related fields, the problem of the contribution of the intrinsic mechanism between PolSAR polarization observation variables to the estimation of forest canopy height is less considered in the study of estimating large-scale forest canopy height using machine learning methods combined with PolSAR observations. ??????? and more throughout so I would suggest a native English speaker and proof reader be employed to fine tune the English. Also, another example: Among the eight polarimetric observables selected, the Free-man-Durden decomposition model is based on the quality of the physical decomposition due to the Freeman-Durden decomposition model. ???? The study area map Figure 1 - looks a little basic for the paper and needs some tidying up re: map.... North Arrow should just be a plain arrow with an N ... not something that looks as if it belongs on a 'Treasure Map'! Why the Red, Green, Blue? No real relevance or purpose. The shaded polygon symbol is the island not the boundary of the island.
In other places the illustrations e.g. graphs/charts could benefit with some annotation to provide an explanation. Some photographs of the forest study site would be useful also. In Figure 6... is it really possible to have a scale showing height as N.XXXm???
Comments on the Quality of English LanguageThe paper is generally well-written and explained - more so than some papers I have read where a complex approach needs to be made accessible to the readership. However, it does need some final editing to remove wordy sentences and sentences that could be better expressed. For example, in the abstract - the first sentence is not the best start! Oriented to the forest carbon stock and carbon cycle related fields, the problem of the contribution of the intrinsic mechanism between PolSAR polarization observation variables to the estimation of forest canopy height is less considered in the study of estimating large-scale forest canopy height using machine learning methods combined with PolSAR observations. ??????? and more throughout so I would suggest a native English speaker and proof reader be employed to fine tune the English. Also, another example: Among the eight polarimetric observables selected, the Free-man-Durden decomposition model is based on the quality of the physical decomposition due to the Freeman-Durden decomposition model. ????
Reviewer 2 Report
Comments and Suggestions for Authors
The use of PoLSAR to estimate the forest cover height is an interesting topic. Overall, this study helps to understanding the direct and indirect contributions of the eight polarization observation variables in the forest canopy height inversion task based on structural equation modelling. The idea and style of the paper give positive impression. The result has some meaning. However, the manuscript has a lot points need be improved.
The description of mathematical equations is excessive. Most of them are integrated into the software programs and machine learning products. Therefore, there is no point in describing them in detail for readers of the Forests. The authors should pay more attention to the description of algorithms to fulfill the assigned tasks and results of the assessments/modelling. It is advisable to adjust the structure of the article, strengthening the methodology, which is often repeated in the results.
Other comments, suggestions:
.Line 41 dynamic manner [1]. More references are needed to show the relevance of the research area.
Line 41-45 The traditional method….. This comparison is an obvious fact that does not require mention in the introduction.
48 on the ground [5] Add several references.
53 range of surface areas. Add references.
56 surface areas. Add references
56 Tasks. Delete.
81-85. Rephrase. Difficult to read and understand.
92 [24,25]
111-119 Not sure that this part is worth including in the text.
125 ….season. Add reference
128 is about 0-40 m. Add reference.
130 Figure 1a Gaofen-3 image… More info needed about this image.
133 Sentinel-2 image obtained on January 2, 2021. Reference to the ESA database needed.
134 we 133 randomly divided the training and test sets in the ratio of 0.9:0.1 for training and accuracy 134 evaluation of forest canopy height estimation in the region.
Accuracy assessment usually requires at least 30% of the test sets.
136 Figure 1. Revise the text according to the a and b descriptions. Eg. Study area on: a) GF-3 image; b) Sentinel-2
What is aim to show RGB bands in b? There will be enough one designation of cardinal directions on the Figure 1.
139 by a laser pulse. Add reference explaining the technical characteristics of this Lidar.
140-142 In this study, the relative canopy height RH98 (h_canopy) was chosen as the reference value of forest canopy height, and h_canopy is 98% of the relative height of the canopy [30-32].
Please clarify the h-canopy or rephrase.
147-153 This paragraph better to include before the figure 1.
147 ICESat-2 LiDAR Reference to the NASA needed.
14-153. This sentence is not entirely clear. Needs clarification or paraphrasing.
184 There is no explanation for the equation 6 in the text. What is the RVI?
(.)∗ Couldn’t find it in the equation.
185 Live some space between the equation and text.
200 (Freeman and Durden , 1998) Delete.
202-206 The RVI equation should be placed here.
187 Provide the Table 2 below this sentence.
249-252. Too much “variables” in the text. Rephrase.
252-256. Divide it into two sentences.
269-274 Should be placed under the Equation 7.
272-274 The weight calculation module ….. calculates the weights. Revise.
288 by Kennedy and Eberhart [56] Direct reference needed.
303-310 There is a duplication of the text given previously. Eg. 133-135, 206-208.
312-338 This text belongs to the methodology part of the manuscript. Some sentences repeat the text above and need to be optimized.
339, 343. Table 3?
349-352 Rephrase
353 There is no description of C, epsilon, and gamma un the text above.
456 remove bold font.
465-469 In this study, we selected….in the study area. Revise.
472 Table 1?
512 Li et al. [26]
529 References should be prepared according to the MDPI requirements.
Comments on the Quality of English LanguageModerate editing of English language required.
Reviewer 3 Report
Comments and Suggestions for Authors
It is important research topic for future about GST (biomass estimation). However, since in my knowledge, C-band can not be penetrated to tree leaf especially high dense area. In addition, there is no geospatial resolution about GF-3 QPS1 and it makes difficult for me to understand the accuracy discussion in your research paper. So, at first, I recommend you to add table to describe data specific in your research paper including GF-3, ICEsat, etc.
In line 214, SEM is mentioned and then you spell out later in line 224. It should be spell out in line 214.
In chapter 3.2, you propose to use PSO algorithm. But, I wonder you have any comparison result with other algorithms. To justify the value of PSO algorithm, I think it is better to mention the advantages of PSO with citation paper more clearly.
In chapter 4, you propose 553 sampling points. Why is it only 553? It maybe also related with geospatial resolution and test site size. If so, you should clearly explain why 553 sampling points and if possible, you should also show the results with less sampling points results.
In Line 318, what is RVI?
In figure 5, big numbers of plots are above dot line like as estimate 12-16. It is more number than result in figure 4.
In discussion, you mention L and X, I recommend you should try to use ALOS data for your comparison study with GF-3 and may not need to do X-SAR because it is more quickly saturated.
Comments on the Quality of English LanguageNA
Reviewer 4 Report
Comments and Suggestions for Authors
The manuscript discusses an approach to estimating forest height using multivariate analysis of features extracted from Gaofen-3 satellite data. The topic of the article is relevant. The authors use modern approaches. At the same time, a number of corrections must be made to the manuscript.
1) Line 16. The abbreviation of the satellite name “GF-3” must be deciphered.
2) Line 20. The designations of the variables T11, T22, T33 must be given with a brief description.
3) Line 56. At the end of the paragraph there is an extra word “Tasks”.
4) Figure 1(b) – Band designations (Red band, Green band, Blue band) in the figure legend must be removed, because The figure shows a multispectral image rather than individual bands.
5) Line 137. The abbreviation “ATL08” must be deciphered.
6) Table 1. In the column title “Resolution” you must indicate the unit of measurement.
7) Line 225. It is recommended to decipher the abbreviation SEM, because it is given at the beginning of the section.
8) Lines 370-371. In the text accompanying the figure 4, it is necessary to provide a short name of the method that was proposed in the study.
9) Figures 4-5. Along the vertical and horizontal axes of the graphs, it is necessary to indicate the unit of measurement for forest height.
10) Lines 385-386. In the text to Figure 6 it is necessary to provide in more detail the name of the methods used.
11) Lines 388-393. It is recommended to delete this part of the text, as a more detailed description is provided later in the text.
12) Section 5.1. Based on Figure 7, the Pearson correlation coefficient is quite low for all variables. A more detailed discussion of the reasons for the low correlation is necessary. What is the statistical significance of the calculated correlations?
13) Figure 7. It is necessary to provide a signature along the vertical axis of the figure. The text for Figure 7 should briefly describe all the variables represented on the horizontal axis.
14) Line 472. You need to check the Table number.
Comments on the Quality of English LanguageThe manuscript should be checked for close repetitions, for example, Line 158: “The geocoded images were then geocoded.”
It is necessary to check the correctness of the definitions used. For example: in the text to Figure 1 (Line 136), instead of “GF-3 image range” it is better to write “Gaofen-3 satellite image extent”.
Round 2
Reviewer 2 Report
Comments and Suggestions for Authors
Dear authors, good to see that you accepted most of my suggestions, which improved the manuscript. Good luck.
Reviewer 3 Report
Comments and Suggestions for Authors
Good to have more citation to previous study for other SAR.
Comments on the Quality of English LanguageNA
Reviewer 4 Report
Comments and Suggestions for Authors
The authors took into account the comments. In this case, it is necessary to make a small clarification in line 154: it is necessary to correct “GF-3 image” to “GF-3 satellite image extent”.
Comments on the Quality of English LanguageIt is recommended to check the manuscript for minor inaccuracies.
